# MicroRNA regulation of murine trophoblast stem cell self-renewal and differentiation

Sarbani Saha*, Rupasri Ain

**Proper placentation is fundamental to successful pregnancy. Placenta arises from differentiation of trophoblast stem (TS) cells during development. Despite being recognized as the counterpart of ES cells in placental development, the role of regulatory miRNAs in TS cell differentiation remains inadequately explored. Here, we have identified complete repertoire of microRNAs present in mouse trophoblast cells in proliferative and differentiated state. We demonstrated that two miRNA clusters, -290 and -322, displayed reciprocal expression during trophoblast differentiation. Loss of miR-290 cluster members or gain in miR-322 cluster members led to differentiation of TS cells. The trophoblast stemness factor, CDX2, transactivated the miR-290 cluster and *Cyclin D1*. MiR-290 cluster members repressed cell cycle repressors, P21, P27, WEE1, RBL2, and E2F7, in TS cells. MiR-322 cluster members repressed the cell cycle activators, CYCLIN D1, CYCLIN E1, CDC25B, and CDX2, to induce differentiation. Taken together, our findings highlight the importance of posttranscriptional regulation by conserved miRNA clusters that form a regulatory network with CDX2, cell cycle activators, and repressors in equipoising TS cell self-renewal and differentiation.**

## Introduction

In Eutherian mammals, the placenta provides the physiological interface between the mother and the fetus and is the sole regulator of nutrient and oxygen supply to the developing embryo. In addition, the placenta also removes waste products from the maternal–fetal compartment and produces a plethora of cytokines and hormones that not only regulate proper fetal development in utero but also regulate maternal physiology conducive for successful pregnancy progression. Placental functions are primarily executed by various lineages of trophoblast cells, which constitute the main structural component of the placenta (Soares et al, 1996; Roberts et al, 2004; Cross, 2005). The precursor of these differentiated trophoblast lineages is multipotent trophoblast stem (TS) cells, which originate from the trophectoderm layer of the

blastocyst (Tanaka et al, 1998). Defective trophectoderm specification results in improper implantation of the embryo (Rossant & Cross, 2001; Cockburn & Rossant, 2010), which is one of the leading causes of early pregnancy failure. After embryo implantation, disruptions in the development and function of trophoblast cells result in pregnancy-related disorders such as intrauterine growth retardation/restriction and preeclampsia (Rossant & Cross, 2001; Perez-Garcia et al, 2018; Woods et al, 2018). Inadequate, trophoblast differentiation leads to adverse pregnancy outcome, which is a global health concern. Thus, an in-depth understanding of the regulatory machinery that controls the development, differentiation and function of the trophoblast cells are critical for the improvement of the infertility treatment and the betterment of reproductive health. Murine TS cells (Tanaka et al, 1998) are an excellent model to analyze molecular regulation of trophoblast cell differentiation ex vivo.

Transcription factors, which regulate the multipotent state of murine TS cells or promote their differentiation, have been well studied (Cross et al, 1995; Kraut et al, 1998; Anson-Cartwright et al, 2000; Scott et al, 2000; Hughes et al, 2004; Knott & Paul, 2014; Baines & Renaud, 2017). Caudal-type homeobox protein-2 (CDX2) is one of the most important stemness-determining factors (Strumpf et al, 2005). Induction of differentiation is associated with robust down-regulation of CDX2, indicating that it is an important regulator of TS cell self-renewal. The role of cell cycle regulators in TS cell maintenance and differentiation has also been studied by several groups. For example, cyclin E–CDK2 complex has been shown to promote trophoblast giant cell (TGC) formation by triggering S-phase entry to facilitate the endoreduplication process (Parisi et al, 2003). Differentiation of TS cells led to up-regulation of two Cdk inhibitors, P57 and P21, resulting in blockage of M-phase entry and thus triggering their differentiation into TGCs (Ullah et al, 2008). However, any regulatory network of cell cycle regulators and transcription factors that control TS cell differentiation remains unclear.

MiRNAs constitute a group of small (21–25 nt) endogenous noncoding single-stranded RNAs which regulate gene expression posttranscriptionally by an imperfect pairing of 6–8 nucleotide (seed sequence) with the target mRNAs resulting in target mRNA silencing through translational inhibition or mRNA degradation or

Division of Cell Biology and Physiology, CSIR-Indian Institute of Chemical Biology, Kolkata, India

Correspondence: rupasri@iicb.res.in
*Recipient of pre-doctoral fellowship from the Council of Scientific and Industrial Research, India

deadenylation (Bushati & Cohen, 2007; Winter et al, 2009). In mammals, miRNAs are highly conserved across species (Chen & Rajewsky, 2007). Almost half of all identified miRNAs are intergenic and transcribed from their own promoters, whereas the remaining are intragenic and processed mostly from introns. Some miRNAs are transcribed as one long transcript called clusters. Biogenesis of miRNAs involves transcription of primary miRNA (pri-miRNA) by RNA polymerase II or III followed by processing into a short hairpin intermediate, called precursor miRNA (pre-miRNA), by the microprocessor complex consisting of an RNA-binding protein, DiGeorge syndrome critical region 8 (DGCR8) and a ribonuclease III enzyme, DROSHA. The pre-miRNA is then exported from the nucleus into the cytoplasm via the nuclear RAN-GTP–dependent exportin protein, EXPORTIN-5. In the cytoplasm, the pre-miRNA is further processed into the mature miRNA duplex by another ribonuclease, DICER, along with the double-stranded RNA-binding protein, TRBP. From this mature duplex, only the functional strand (guide strand) becomes incorporated into the RNA-induced silencing complex (RISC) containing the RNA processing protein Argonaute (AGO), whereas the nonfunctional passenger strand is degraded. Now, the guide strand aids the binding of the RISC to the target mRNA using its 5′ seed sequence, which is complementary to the 3′ UTR of target mRNA, leading to posttranscriptional silencing of mRNA targets through mRNA cleavage, translational repression, or deadenylation.

The first evidence for the role of miRNA in TS cell development was established by Spruce et al (2010). Their attempt to establish TS cell line from Dicer null mice failed. Furthermore, down-regulation of Dicer in TS cell line, established from the conditional Dicer knockout mouse, resulted in loss of TS cell phenotype, leading to their differentiation into giant cells. These results clearly indicated the role of miRNAs in TS cell maintenance. Some members of the miRNA cluster C19MC on chromosome 19 was detected in the human term placenta (Bortolin-Cavaillé et al, 2009) and in maternal blood (Miura et al, 2010). Furthermore, miRNA expression profiling was carried out in human immortalized extra-villous first trimester trophoblast cell line HTR-8/SVneo, the choriocarcinoma cell line JEG-3, and primary trophoblast cells isolated from the first trimester and term human placenta (Morales-Prieto et al, 2012). This study provides a comprehensive encyclopedia of the miRNA expression profile of human trophoblast cell lines widely used as models of trophoblast cells, and their comparison with primary isolated trophoblast cells from the first and third trimesters. However, numerous dissimilarities were reported in this data set. Although miRNAs are likely to play a central role in regulatory pathways controlling lineage determination, cell differentiation, and function of trophoblast cells, the impact of miRNA-mediated regulation on the TS cell self-renewal and differentiation remains largely unknown.

In this report, we identified a compendium of miRNAs that are differentially regulated during TS cell differentiation. We demonstrated the mechanism by which miRNA clusters miR-290 and miR-322 regulate maintenance of stemness in trophoblast cells and their differentiation. Our data unveiled a regulatory network of lineage-determinant transcription factor, CDX2, and cell cycle regulators with miRNA clusters, which equipoise TS cell self-renewal and differentiation.

# Results

## MiRNA PCR array analysis reveals importance of miRNA clusters in TS cell self-renewal and differentiation

The emerging role of miRNAs in development and differentiation as well as failure to establish Dicer-null stable TS cell line (Spruce et al, 2010) prompted us to investigate the regulatory role of miRNAs in TS cell self-renewal and differentiation. The first step in this quest was to analyze the complete repertoire of 940 best characterized mouse miRNAs in TS cells and differentiated trophoblast cells ex vivo (Fig S1A) using miRNome miScript miRNA PCR array. Overall, 169 miRNAs met the recommended cutoff reads (Ct ≤ 30) in at least one of the two groups and changed significantly (≥2-fold, P < 0.05) between two groups in the array (Fig 1A). Of these, 94 miRNAs were more abundant in TS cells and were down-regulated upon differentiation (Table S1), whereas 75 miRNAs were poorly expressed in TS cells and were up-regulated in differentiated trophoblast cells (Table S2).

From these differentially expressed miRNAs, two clustered miRNA groups, miR-290 and miR-322 clusters, were identified using miRNA database, miRBase (Fig 1B and Table S3). Interestingly, some members of the miR-290 cluster have been previously reported to regulate the cell cycle repressor, P21, in ES cells. However, there was no such report on the miR-322 cluster. Analysis of these two cluster members by using various target prediction tools, such as TargetScan, PicTar, and miRNA.org, showed that members of the miR-290 cluster are predicted to regulate cell cycle repressors, whereas miR-322 cluster members are predicted to regulate cell cycle activators. The star or passenger strands of miRNA members of these two clusters and the members which do not have any relevant cell cycle regulator as their target were excluded from this study (Table S3, only the miRNAs written in bold were selected for further study).

Differential expression of selected miRNAs from these two clusters was further validated by TaqMan assay using U6 snRNA as an endogenous control. In line with the microarray data, miR-290 members, miR-291a-5p, miR-291b-3p, miR-292a-3p, miR-294- 3p, and miR-295-3p, were highly abundant in TS cells and were down-regulated in differentiated cells (Fig 1C). On the contrary, miR-322 members, miR-322-5p, miR-503- 5p, miR-351-5p, miR-542-3p, and miR-450b-5p, were expressed highly upon induction of differentiation (Fig 1D). Furthermore, expression of two representative miRNAs from each cluster was assessed on day 2, day 4, and day 6 of differentiation in trophoblast cells (Fig S1B). A gradual temporal decrease in miR-290 cluster members, miR-291b-3p and miR-295-3p, was observed with progression of differentiation. MiR-322 cluster members, miR-322-5p and miR-503-5p, expressed at considerably high levels upon induction of differentiation on day 2 and day 4. However, a robust up-regulation was observed on day 6 of differentiation.

## MiR-290 cluster potentiates TS cell self-renewal by targeting cell cycle repressors

In silico target prediction revealed seven cell cycle repressors, P21, P27, WEE1, RB1, RBL1, RBL2, and E2F7, as targets for miR-290 cluster members (Fig S2A). However, real-time PCR analysis of these transcripts in TS and differentiated cells showed that *Rb1* and *Rbl1*

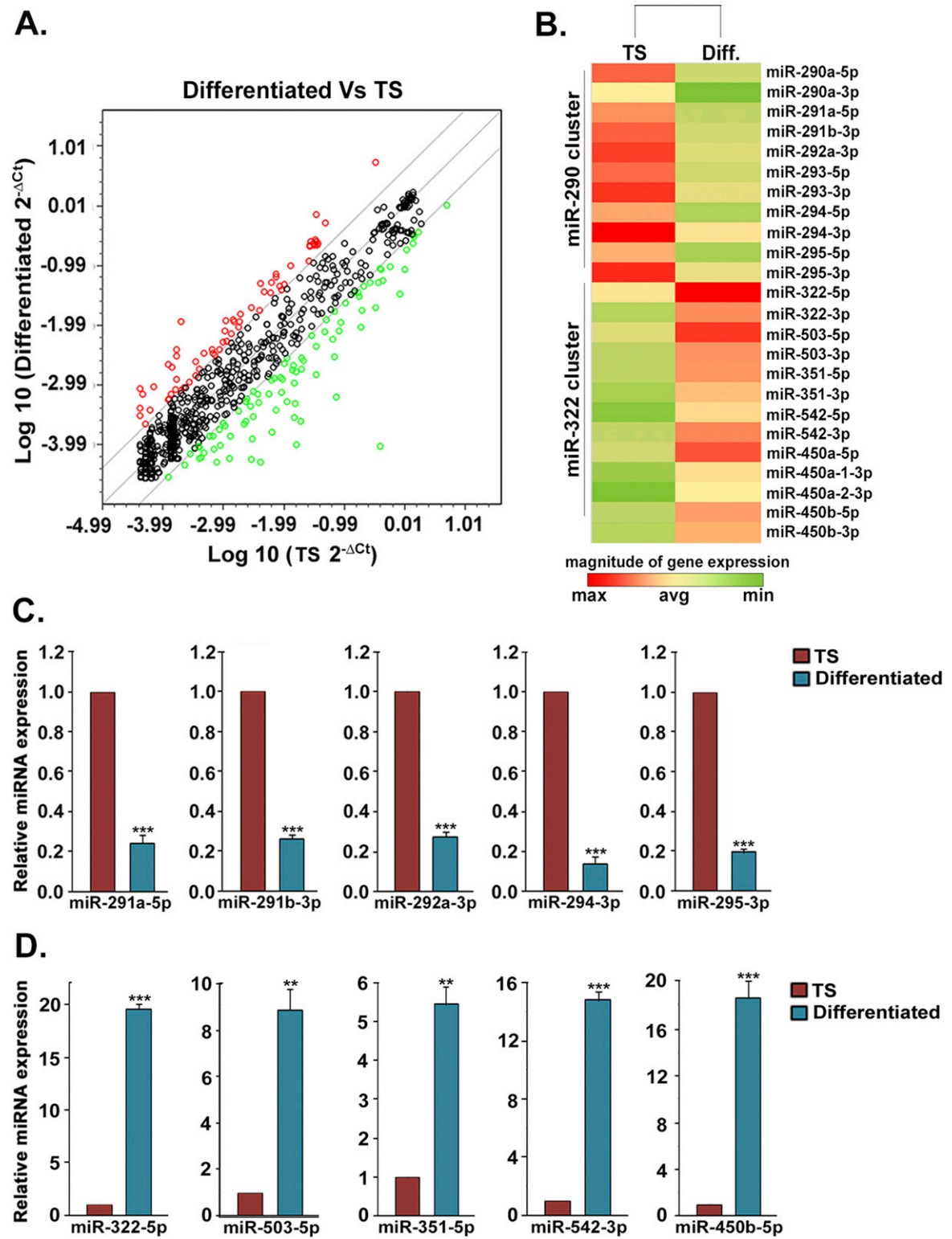

**Figure 1. MiRNome PCR array profiling of miRNAs in trophoblast stem (TS) cells and differentiated trophoblast cells.**
**(A)** Scatter plot representing differential expression of 169 miRNAs, of which 94 miRNAs were down-regulated (green) and 75 up-regulated (red) in differentiated trophoblast cells. **(B)** Clustergram for differential expression of miR-290 and miR-322 clusters in TS cells and differentiated cells. **(C, D)** TaqMan assays for the members of miR-290 and miR-322 clusters in TS cells and differentiated cells. Bars represent the mean ± standard error of the mean of three independent experiments (n = 3). **P < 0.005; ***P < 0.0005 when compared with TS cells.
Source data are available for this figure.

were not functionally relevant in the context of trophoblast differentiation (Fig S2B). Protein levels of P21, P27, WEE1, RBL2, and E2F7 were in concordance with their transcript levels and were found to be up-regulated in differentiated cells (Fig S2C).

To explore whether the miR-290 cluster directly regulates the aforesaid cell cycle repressors and controls, TS cell self-renewal, gain in function, or loss of function studies were performed by

transfecting mimics and inhibitors for miR-290 cluster members into TS cells. TS cells, transfected with mimics of respective miRNAs, exhibited decreased expression of the cell cycle repressors P21, P27, WEE1, E2F7, and RBL2 at both mRNA (Fig 2A) and protein (Fig 2B) levels as compared with scrambled transfected controls. Inhibitors of miR-290 cluster members enhanced the expression of these cell cycle repressors at both mRNA and protein levels in TS cells (Fig 2A

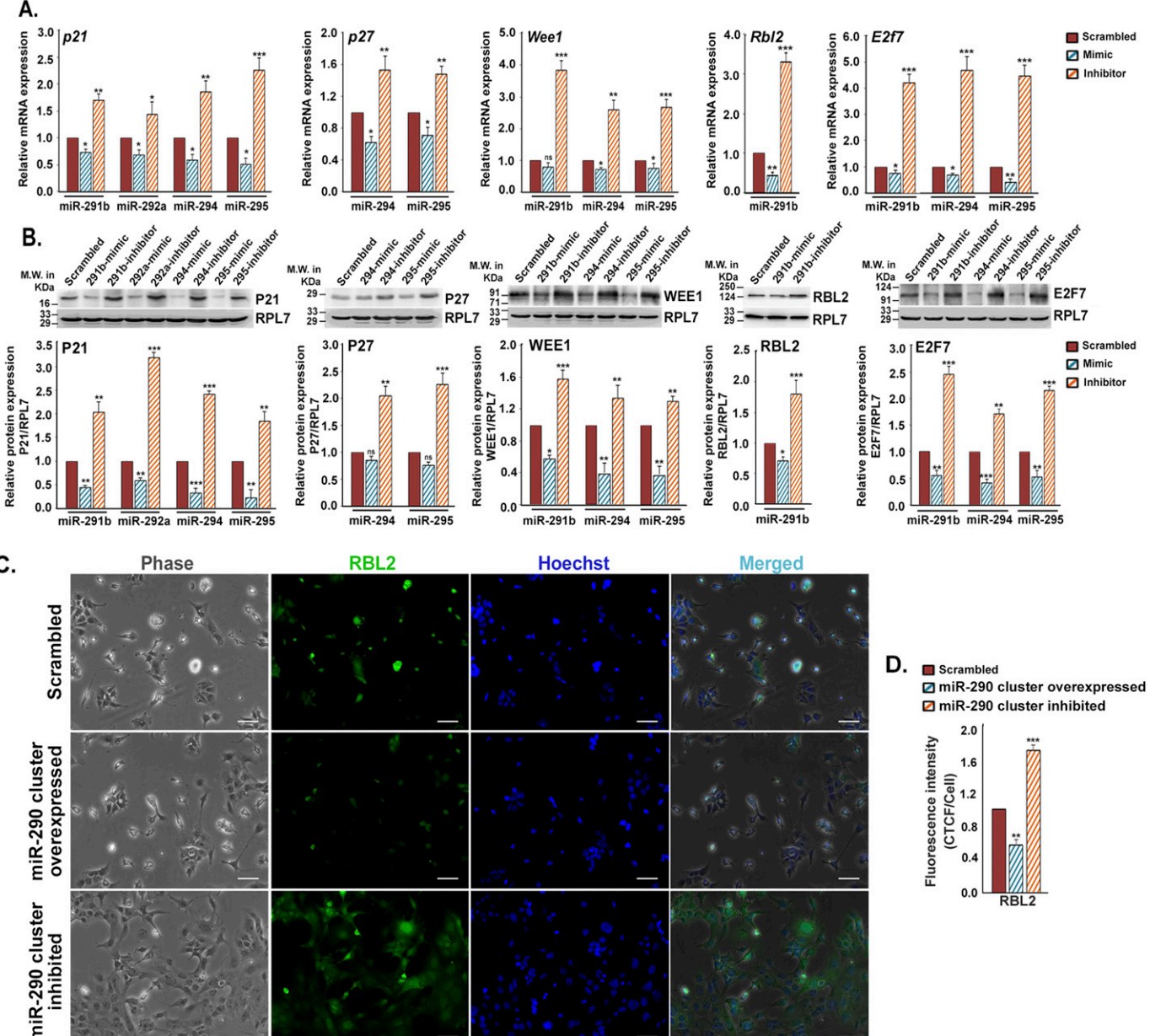

**Figure 2.   Members of miR-290 cluster target cell cycle repressors to maintain stemness.**
**(A)** Quantitative real-time PCR of cell cycle repressors in trophoblast stem (TS) cells transfected with either mimic or inhibitor alone for members of miR-290 cluster. **(B)** Immunoblot analysis of cell cycle repressors in TS cells transfected with either mimic or inhibitor alone for members of miR-290 cluster. RPL7 was used as loading control. Densitometric quantification of each protein sample relative to RPL7 is shown in the following. Data are presented in means and standard error of the mean of three replicates (n = 3). **(C)** Immunofluorescence of RBL2 (green) in TS cells transfected either with all the mimics or all the inhibitors of miR-290 cluster showing the same results. Cell nuclei were stained with Hoechst dye (blue). Scale bar: 50 μm. **(D)** Quantification of fluorescence intensity for RBL2 from C, panel 2. Values are shown in mean ± SEM from three different experiments (n = 3). *P < 0.05; **P < 0.005; ***P < 0.0005; ns, nonsignificant.
Source data are available for this figure.

and B). Further confirmation of these findings was carried out by immunofluorescence staining. One representative example is shown in Fig 2C. RBL2 is immunostained in TS cells, transfected either with all the mimics or all the inhibitors of the miR-290 cluster (Fig 2C), and scrambled transfected cells were used as control. Interestingly, inhibitor-mediated down-regulation of this cluster was not only associated with increased expression of RBL2 (Fig 2D) but also showed induction of differentiation in TS cell (Fig 2C), as observed by change in morphology.

### MiR-322 cluster induces differentiation in mouse TS cell by targeting cell cycle activators

As all the six members of the miR-322 cluster showed remarkable up-regulation in differentiated trophoblast cells, the potential role of the miR-322 cluster on the induction of differentiation of mouse TS cells was examined. The genes, predicted to be targets of miR-322 cluster members by in silico target analysis, are primarily various cell cycle–promoting factors (Fig S3A). Expression analysis revealed that of the eight predicted targets, only three, CYCLIN D1, CYCLIN E1, and CDC25B, were relevant in the context of TS cell differentiation (Fig S3B and C). To assess the function of the miR-322 cluster on trophoblast differentiation, gain in function or loss of function studies using mimics and inhibitors of the cluster members was performed in TS cells. RNA and protein levels of the cell cycle activators were analyzed. Mimic of both miR-322-5p and miR-503-5p robustly down-regulated *cyclin D1* transcripts and proteins. As expected, effects of the inhibitors were rather modest because of low levels of expression of these miRNAs in TS cells (Fig 3A and B). In contrast to these findings, miR-322/miR-503 mimics did not curtail the expression of *Cyclin E1* so prominently (Fig 3A and B), indicating existence of other regulatory mechanisms for expression of this cyclin. Transfection of miR-322-5p mimic in TS cells resulted in significant down-regulation of CDC25B, while its expression was found to be increased upon addition of the miR-322-5p inhibitor (Fig 3A and B). Similar to miR-290 cluster inhibition, ectopic over-expression of all the members of miR-322 cluster induced the differentiation of TS cells (Fig 3C, middle panel), which coincided with significant down-regulation of the immunofluorescence staining of CYCLIN D1 in those differentiated TS cells (Fig 3C middle panel and Fig 3D). Altogether, these data clearly demonstrated the role of the miR-322 cluster in suppression of cell cycle activators in TS cells, which might induce the differentiation process.

### MiR-290 and -322 clusters affect cell proliferation and differentiation by altering the expression of signature transcription factors of TS cells

To test whether miR-290 and miR-322 clusters directly affect cell proliferation and differentiation of TS cells, BrdU incorporation and transcript levels of genetic markers for TS cells and differentiated trophoblast cells were analyzed. Inhibition of all the potential members of miR-290 clusters led to 50% reduction in BrdU incorporation in TS cells (Fig 4A) associated with a decrease in CDX2 expression (Fig 4B and C). Overexpression of all the potential members of miR-322 clusters in TS cells yielded similar effects (Fig 4A–C). Careful observation of morphology of cells in the BrdU incorporation assay indicated that the decrease in BrdU incorporation

was attributed to induction of differentiation resulting from miR-290 cluster inhibition or miR-322 cluster overexpression. TS cell lineage–determining transcription factor, *Cdx2*, and other TS cell signature transcription factors, such as *Eomes* and *Esrrb*, were down-regulated by inhibition of the miR-290 cluster or overexpression of the miR-322 cluster (Fig 4D–G). As expected, early differentiation markers, *Plf* and *Pl1*, were up-regulated in both instances (Fig 4D–G).

Next, to compare the differentiation induced by miR-290 cluster inhibition or miR-322 overexpression with differentiation by withdrawal of mitogens (MEF-CM, FGF4, and heparin), TS cells were transfected either with all the inhibitors of miR-290-cluster or all the mimics of the miR-322 cluster and transfected cells were maintained in stemness-maintaining conditions. Besides, differentiation was induced in TS cells by withdrawal of MEF-CM, FGF4, and heparin. 72 h following transfection, RNA was isolated from the transfected cells. Differentiated trophoblast cell phenotype was assessed using genetic markers for various lineages. *Plf*, *Pl1*, and *Ctsq* were used as markers for TGCs; *Mash2* and *Tpbpα* were used as spongiotrophoblast marker; and *Gcm1* was used as syncytiotrophoblast marker. Extent of differentiation by miR-290 inhibition or miR-322 overexpression was relatively low as compared with differentiation by mitogens withdrawal (Fig S4).

### CDX2 regulates self-renewal by trans-activating the miR-290 cluster and cyclin D1

CDX2 is critical for establishment of mouse TS cells (Strumpf et al, 2005), and its depletion causes spontaneous differentiation of TS cells (Saha and Ain, unpublished data). It is exclusively expressed in TS cells and is turned off upon differentiation (Fig S5A and B). Abundance of the miR-290 cluster in TS cells led us to analyze putative CDX2-binding sites in the promoter region of the miR-290 cluster and the cell cycle activators using a computational promoter analysis approach. Interestingly, multiple CDX2-binding motif TTTAT (Amin et al, 2016) was identified on the promoter regions of the miR-290 cluster and cyclin D1 (Fig 5A and B). Chromatin immunoprecipitation (ChIP) using CDX2 antibody followed by PCR analysis using the TS cell nuclear extract revealed that CDX2 binds to all the three sites of the miR-290 cluster promoter in which binding site 2 (BS2) showed the highest binding intensity. In addition, ChIP assay using RNA pol-II antibody and PCR confirmed that CDX2-bound sites on the miR-290 cluster were transcriptionally active (Fig 5A). Similar ChIP PCR analysis in TS cells confirmed that CDX2 transactivates the cyclin D1 promoter in TS cells (Fig 5B).

To analyze the importance of the miR-290 cluster and cyclin D1 promoter occupancy by CDX2 in TS cells, CDX2 was either down-regulated using siRNAs or overexpressed in TS cells. Down-regulation of Cdx2 was confirmed using real-time PCR (Fig S5C) and Western blotting (Fig S5E and F). *Cdx2* down-regulation curtailed *cyclin D1* transcript and protein expression (Fig 5C and D). TaqMan assay revealed that down-regulation of *Cdx2* also abrogated expression of miR-290 cluster members in TS cells (Fig 5E). In addition, *Cdx2* down-regulation resulted in up-regulation of cell cycle repressors (P21, P27, WEE1, RBL2, and E2F7), transcripts, and proteins (Fig S5D, E, and G). Ectopic overexpression of CDX2 in TS cells (Fig S6A, C, and D) led to up-regulation of miR-290 cluster members (Fig 5F) associated with down-regulation of cell cycle repressors (Fig S6B, C, and E).

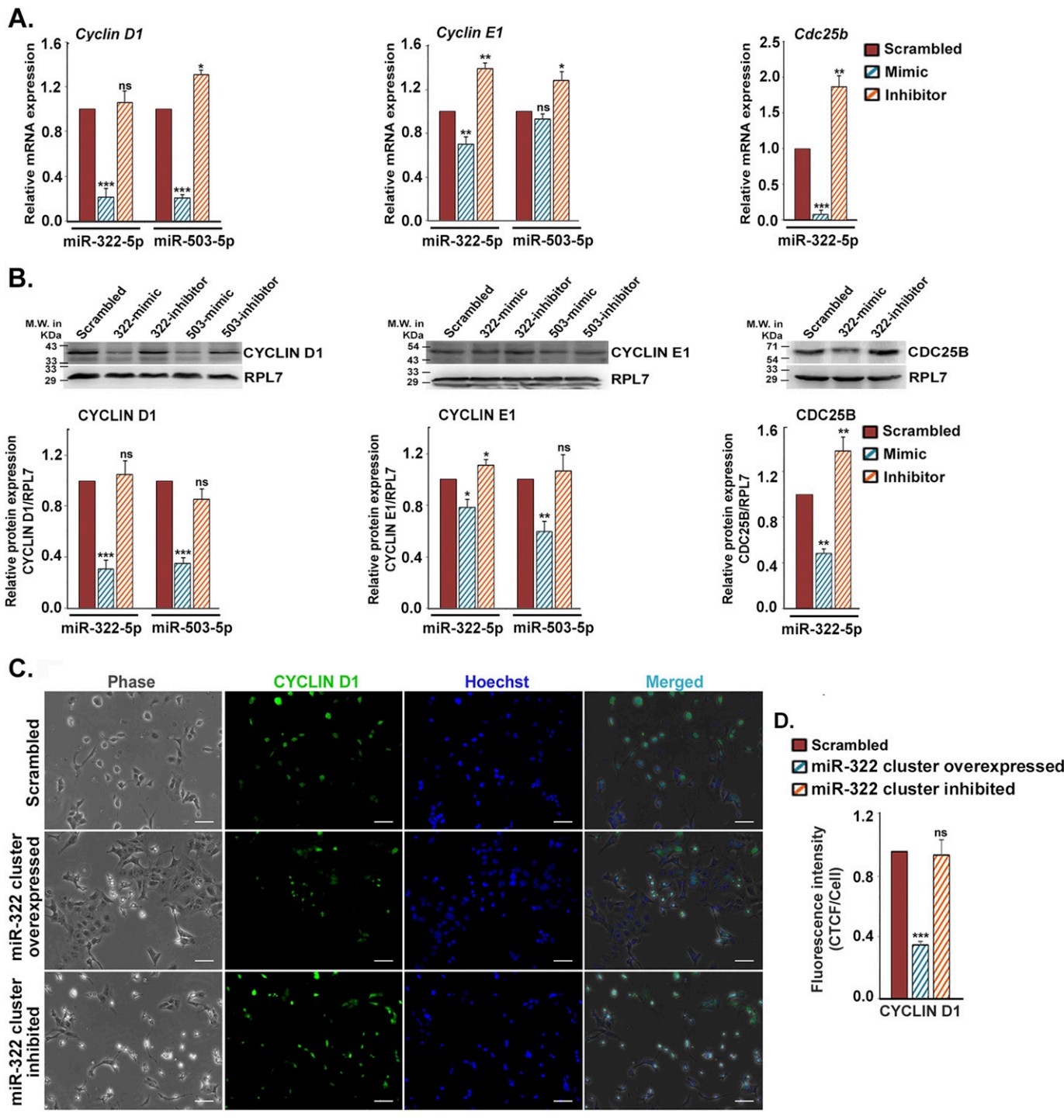

**Figure 3. MiR-322 cluster induces mouse trophoblast stem (TS) cell differentiation by targeting cell cycle activators.**
**(A)** Quantitative real-time PCR of cell cycle activators (*Cyclin D1*, *Cyclin E1* and *Cdc25b*) in TS cells transfected with either mimic or inhibitor for miR-322-5p and miR-503-5p. **(B)** Western blot analysis of these activators from transfected TS cells. RPL7 was used as loading control. Densitometric quantification of every protein sample relative to RPL7 is shown in the following. Data are presented in means and standard error of the mean of three replicates (n = 3). **(C)** Immunofluorescence of CYCLIN D1 (green) in TS cells transfected with all the mimics or all the inhibitors of miR-322 cluster. Scale bar: 50 μm. **(D)** Quantification of fluorescence intensity for CYCLIN D1 from C, panel 2. *P < 0.05; **P < 0.005; ***P < 0.0005; ns, nonsignificant. Source data are available for this figure.

Furthermore, to assess whether ectopic overexpression of CDX2 was able to reverse the effect of miR-290 cluster inhibition or miR-322 cluster overexpression in TS cells, miR-290 cluster inhibition or miR-322 cluster overexpression was carried out along with CDX2 overexpression. Genetic markers for TS cells (*Eomes* and *Esrrb*) and differentiated trophoblast cells (*Plf* and *Pl1*) were used to assess

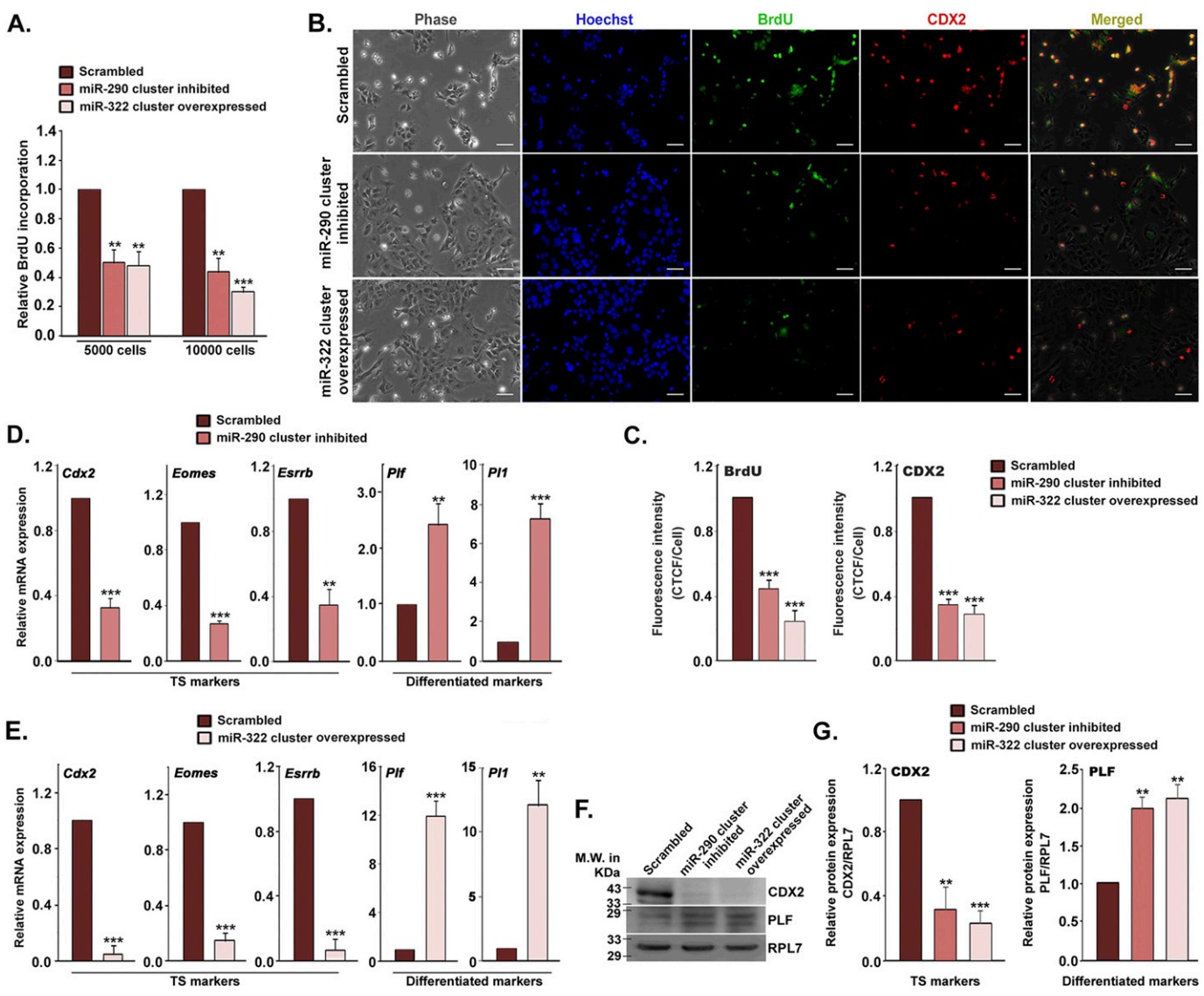

**Figure 4. Mir-290 and -322 cluster impact cell proliferation and differentiation of trophoblast stem (TS) cells.**
**(A)** BrdU cell proliferation assay in TS cells treated with miR-290 cluster inhibitors or miR-322 cluster mimics. Values are the mean ± SEM of four independent determinations (n = 4). **(B)** Immunofluorescence of BrdU (green) incorporation, CDX2 (red) in TS cells inhibited with miR-290 cluster or overexpressed miR-322 cluster. Scale bar: 50 μm. **(C)** Quantification of fluorescence intensity for BrdU and CDX2 from B, panels 3 and 4 (n = 3). **(D, E)** Real-time PCR analysis of stemness markers (*Cdx2*, *Eomes*, and *Esrrb*) and early TGC marker (*Plf* and *Pl1*) in TS cells treated with miR-290 cluster inhibitors (D) or miR-322 cluster mimics (E). **(F)** Western blot analysis of CDX2 and PLF in TS cells treated with miR-290 cluster inhibitors or miR-322 cluster mimics. **(G)** Quantification of protein expression using NIH imageJ software. **P < 0.005; ***P < 0.0005 compared with control. Source data are available for this figure.

the self-renewing state and differentiated state, respectively. Results from these experiments demonstrate that CDX2 overexpression can reverse the effect of miR-290 cluster inhibition or miR-322 cluster overexpression (Fig S7).

### MiR-322 cluster members directly inhibit lineage-determining stemness factor CDX2

The transcription factor, CDX2, is the key regulator of trophoblast lineage determination and trophoblast stemness maintenance. Our data (Fig 4E) indicated posttranscriptional regulation of *Cdx2* in TS cells. These results led to further in silico analysis using various target prediction software, and it was found that 3′UTR of

mouse *Cdx2* mRNA harbors binding sites for three miR-322 cluster members, miR-322- 5p, miR-503-5p, and miR-542-3p (Fig S8). To validate *Cdx2* as a target of multiple miRNAs (miR-322-5p, miR-503-5p, and miR-542-3p), dual luciferase assay was performed. The dual luciferase reporter construct contained firefly luciferase cDNA fused to the 3′UTR of *Cdx2* containing either the binding sites of the microRNAs or miRNA-binding sites with two point mutations in the seed region (Fig S8). Reduction of luciferase activity was observed by transfection of miR-322-5p, miR-503-5p, or miR-542-3p mimics as compared with scramble (Fig 5G). Specificity of the inhibition of luciferase activity was demonstrated by introducing point mutation in the seed region of the miRNA-binding sites in the cloned 3′-UTR of *Cdx2* (Fig S8). Mutated

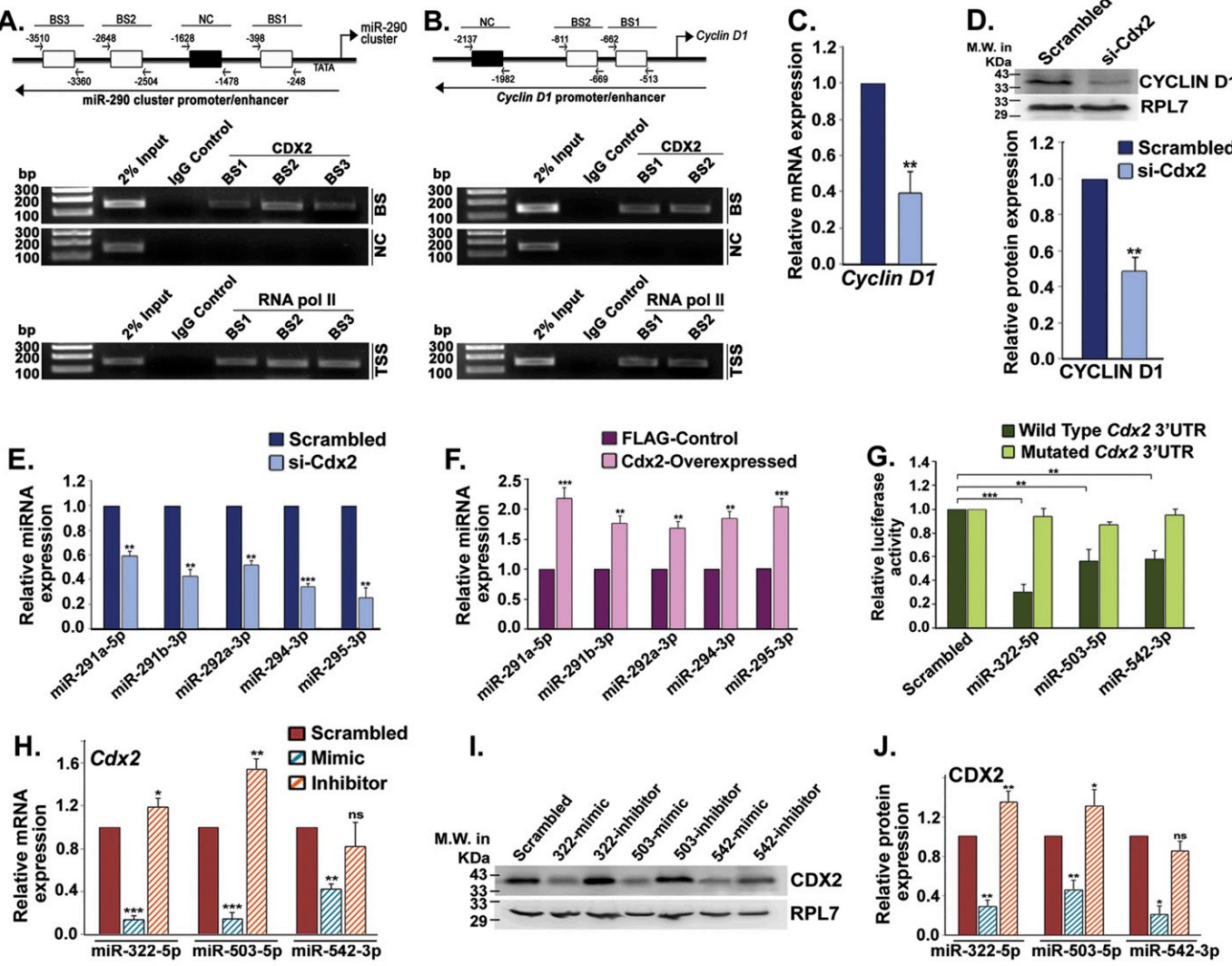

**Figure 5. CDX2 transactivates miR-290 cluster, cyclin D1 promoter in trophoblast stem (TS) cells, and miR-322 cluster members directly inhibit CDX2.**
**(A, B)** ChIP analyses revealed direct binding of CDX2 in miR-290 cluster (A) and cyclin D1 (B) promoter regions. BS indicates binding site. **(C, D)** Real-time PCR analysis of *Cyclin D1* (C) and Western blot analysis of CYCLIN D1 (D) in CDX2 knocked down TS cells. Shown in the following is the NIH imageJ analysis of Western blot. **(E)** TaqMan assays of miR-290 cluster members in Cdx2 knocked down TS cells. **(F)** TaqMan assays of miR-290 cluster members in CDX2 overexpressed TS cells. **(G)** Relative luciferase assay showing the repressive effect of miR-322/503/542 mimic on wild-type Cdx2-3′-UTR. **(H, I, J)** Quantitative real-time PCR analysis of *Cdx2* (H) and Western blot analysis of CDX2. **(I)** in TS cells transfected with mimic or inhibitor of miR-322 cluster members. Quantification of protein expression using NIH imageJ analysis of the Western blot. **(J)** Data are presented in means and standard error of the mean of three replicates (n = 3). *$P < 0.05$; **$P < 0.005$; ***$P < 0.0005$; ns, nonsignificant compared with control. Source data are available for this figure.

3′UTRs failed to inhibit luciferase activity (Fig 5G). To understand physiological relevance of these findings, the miRNA mimics or inhibitors were transfected into TS cells and levels of *Cdx2* transcripts and protein were measured. Compared with scramble transfected controls, both CDX2 mRNA and protein expression significantly decreased in miR-322 or miR-503 or miR-542 overexpressing TS cells (Fig 5H–J), although it remained almost unaffected in TS cells transfected with miR-322/miR- 503/542 inhibitors (Fig 5H–J).

A summary of our data has been shown in Fig 6. In TS cells, CDX2 transactivates miR-290 cluster and cyclin D1. MiR-290 cluster members suppress the cell cycle repressors. Elevated levels of CYCLIN D1 and decreased expression of cell cycle repressors promote proliferation of TS cells resulting in self-renewal. Differentiation of TS cells results in up-

regulation of miR-322 cluster members, which suppresses expression of cell cycle activators, *Cyclin D1* and *Cdc25b*. In addition, miR-322 cluster suppresses expression of *Cdx2*. Lack of CDX2 results in the decline of miR-290 cluster members, thus unmasking their suppressive effect on cell cycle repressors. Thus, decreased expression of cell cycle activators and increased expression of cell cycle repressors result in inhibition of TS cell proliferation, leading to differentiation of TS cells.

# Discussion

In the course of evolution, development of placenta is one of the most important mammalian adaptations, which has made the

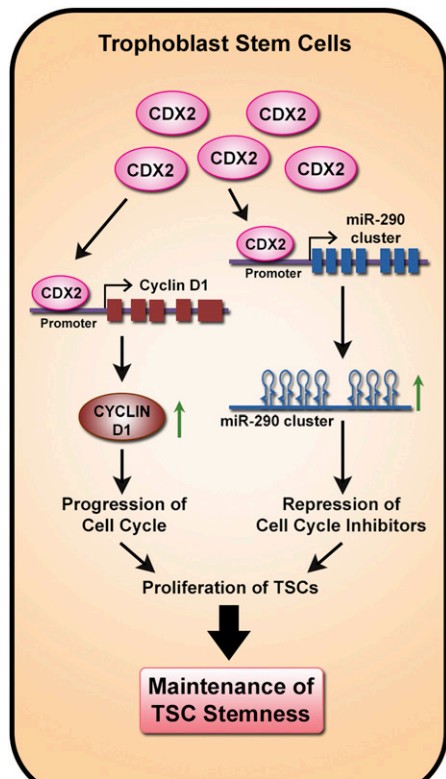
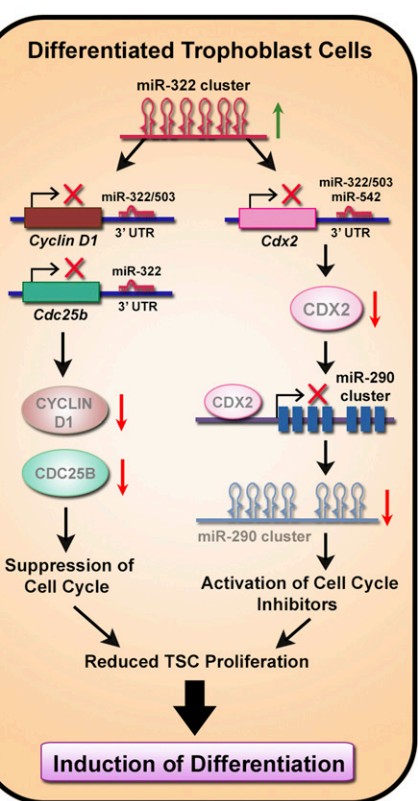

**Figure 6. Schematic representation of the regulatory network formed by CDX2, cell cycle regulators, and miRNA clusters in trophoblast stem (TS) cells and differentiated trophoblast cells.**

CDX2 is abundant in TS cells. It transactivates the miR-290 cluster and cyclin D1 by binding to their promoter regions. Subsequently, the miR-290 cluster members suppress the cell cycle inhibitors (CKIs), whereas CYCLIN D1 promotes the proliferation of TS cells, and thus collectively, maintain the stemness. Upon differentiation, the miR-322 cluster is up-regulated, leading to suppression of CDX2 expression. Depletion of CDX2 leads to down-regulation of the miR-290 cluster and consequent up-regulation of CKIs. Members of the miR-322 cluster also down-regulate the CYCLIN D1 and CDC25B levels to abolish the G1/S transition and suppress the proliferation of TS cells and thus induce TS cell differentiation.

Eutherian reproductive strategy more successful than other species. TS cells, the progenitor cells that give rise to all the other trophoblast subtypes of the placenta, are extensively used as an in vitro model to study the development of the placenta (Soares et al, 1996; Rossant & Cross, 2001; Roberts et al, 2004; Cross, 2005). In the past two decades, a great deal of success has been made toward identifying the signaling pathways and transcription factors that control the trophoblast differentiation event. FGF4-heparin signaling and transcription factors CDX2 along with a plethora of transcription factors were proved to be very important in maintenance of trophoblast stemness (Russ et al, 2000; Strumpf et al, 2005; Baines & Renaud, 2017). By contrast, transcription factors such as HAND1, MASH2, and GCM1 govern trophoblast differentiation (Guillemot et al, 1994; Cross et al, 1995; Scott et al, 2000), which ultimately leads to the formation of the maternal–fetal interface. However, very little information is known about the function of noncoding RNAs in differentiation of TS cells. In this study, we have identified the repertoire of miRNAs expressed in TS cells and differentiated trophoblast cells and also unveiled a regulatory network between noncoding miRNAs; stemness maintenance factor, CDX2; and various cell cycle regulatory proteins.

Of the 940 available and well-characterized miRNAs that were analyzed in this report, 169 miRNAs were differentially regulated during TS cell differentiation. We focused only on two clusters and showed that CDX2 transactivates the miR-290 cluster, which in turn represses the cell cycle repressors and thus enables uninterrupted self-renewal of TS cells. From the levels of expression of these miRNA cluster members in TS cells and differentiated cells (note the Ct values), it appears that regulation by a single miRNA might not be an all or none phenomenon in trophoblast differentiation. Rather, their regulatory cohorts act as a buffer in cellular context to alter gene expression, leading to developmental changes. Besides, a particular cluster might have multiple functions, for which they need to be turned on at a low level even in differentiated cells. Long noncoding RNAs that are known to act as sponges for miRNA also could be regulating their expression levels (Shan et al, 2018). Our data emphasize the regulatory potential of miRNA clusters in TS cell stemness maintenance and differentiation. Gradual rise and fall in expression of the miRNA cluster members with temporal progression of differentiation further emphasize their role in equipoising self-renewal and differentiation of TS cells. The compendium of miRNAs that we generated will provide a valuable start point for other researchers to explore functions of various miRNAs in trophoblast cell function. However, caution might be exercised while working with bioinformatics databases and miRNAs identified by us because as can be seen from our data that all targets found in bioinformatics databases might not be physiologically relevant in trophoblast cells (Figs S2 and S3).

Our data showed that the effects of miR-290 cluster inhibitors were more potent than the mimics on the expression of cell cycle repressors in the TS cells (Fig 2). This might primarily be due to the presence of all the cluster members in TS cells implicating the regulatory effects of the miR-290 cluster on the endogenous expression of these cell cycle repressors. Furthermore, it was also observed that miR-290 cluster inhibitors induced differentiation of TS cells. Therefore, it is likely to synergistically enhance expression of cell cycle repressors along with the effect of the miRNA cluster inhibitors. On the contrary, the effect of miR-322 cluster mimics on

the expression of cell cycle activators was robust as opposed to the corresponding inhibitors in TS cells (Fig 3). This observation is in line with our finding that levels of miR-322 cluster expression in TS cells are quite low, and hence, the effect of inhibition is marginal. Our data on the effect of gain in function or loss of function of miRNA clusters in cell proliferation assay highlight the importance of performing both colorimetric and morphometric analyses. It is evident that the induction of differentiation caused by suppression of the miR-290 cluster or enhanced expression of the miR-322 cluster was the sole cause of decline in BrdU incorporation (Fig 4). These data further affirm the roles of these two miRNA clusters in TS cell self-renewal and differentiation. Interestingly, the nature of differentiation induced by either inhibition of the miR-290 cluster or overexpression of the miR-322 cluster followed the same pattern of differentiation induced by mitogens withdrawal. However, the extent of differentiation was less robust by modulation of a single miRNA cluster, indicating involvement of multiple layers of regulation of TS cell differentiation.

Our results on miR-322 and miR-503 binding to the 3′-UTR of *Cdx2* transcripts is quite interesting as this is the first report demonstrating how the stemness factor CDX2 is posttranscriptionally regulated upon induction of differentiation leading to its degradation (Fig 5). Our data on transactivation of the miR-290 cluster and cyclin D1 by CDX2 are also remarkable as they establish the hitherto unknown regulatory network between CDX2, regulatory miRNA clusters, and cell cycle regulators. TS cells are considered as a developmental counterpart of ES cells in the context of placental development. Interestingly, members of the miR-290 cluster are well-known for their roles in maintaining the self-renewal of mouse ESCs (Wang et al, 2008) and hence previously considered as ESC-specific cell cycle–regulating miRNAs (Judson et al, 2009). Furthermore, Nosi et al (2017) have demonstrated that ectopic overexpression of miR-322 can induce trophoblast-like phenotypes in mouse embryonic stem cells. ES cells are pluripotent, and TS cells are multipotent. Therefore, it is likely that moderate expression of miR-322 (real-time $C_t$ ~ 22.1) leading to down-regulation of cell cycle activators in ES cells will drive differentiation into the default TS pathway. For TS differentiation, robust up-regulation (~16–18-fold) of miR-322 is required, as shown by our data. The existing literature and our data indicate that there is functional conservation of miRNAs in various stem cell types. However, the regulation of their expression might be unique to specific stem cell types.

In conclusion, our results demonstrate that an essential feature of the maintenance and differentiation of TS cells is their acquisition of functional regulatory miRNA repertoire and that their miRNA targets represent key regulatory points in the control of TS cell self-renewal and differentiation. Furthermore, miRNA clusters represent key mediators of the trophoblast cell phenotype. Finally, a complex regulatory network of stemness factors, miRNA clusters, and cell cycle regulators fine-tunes the proliferative and differentiation decisions in the course of trophoblast development.

# Materials and Methods

### Mouse TS cell culture and differentiation

Mouse blastocyst-derived TS cells were a kind gift from Prof. Janet Rossant, Hospital for Sick Children, Toronto, Canada. To maintain the cells in proliferative state, cells were cultured in 30% TS complete media (RPMI-1640 [Sigma-Aldrich] supplemented with 20% FBS [Invitrogen], 1 mM Na-pyruvate [Sigma-Aldrich], 100 μM β-mercaptoethanol [Sigma-Aldrich], 1% GlutaMAX [Invitrogen], and 1% pen–strep [Invitrogen]) supplemented with 70% MEF-conditioned media, FGF4 (R&D Systems), and heparin (Sigma-Aldrich), as described previously (Saha et al, 2015). Differentiation was induced by withdrawal of MEF-conditioned medium, FGF4, and heparin. TS cells were passaged using 0.05% trypsin–EDTA (Invitrogen). Differentiated trophoblast cells were maintained till day 6. Withdrawal of mitogens led to differentiation into various lineages following a specific time course, as described earlier (Saha et al, 2015; Chakraborty et al, 2018, 2020). However, on day 6 of differentiation, most of the cells were primarily TGCs with some population of spongiotrophoblast cells.

To obtain MEF-conditioned medium, MEF cells were isolated from the day 13.5 embryos of C57BL6 mice and cultured using DMEM high-glucose media (Invitrogen) supplemented with 10% FBS, 1% GlutaMAX, 1% nonessential amino acids (Invitrogen), 1% pen–strep, and 0.11 g/ml Na-pyruvate. At 90–95% cell confluence, MEFs were treated with mitomycin C (Sigma-Aldrich) for 3 h and washed with PBS. Cells were then fed with TS complete medium and cultured for 72 h. The conditioned medium was collected in three batches after every 72 h. The conditioned medium was centrifuged and filtered to remove cell debris and stored at −80°C and used for culturing TS cells (Chakraborty & Ain, 2018).

### RNA preparation, reverse transcription, and real-time qPCR

Total cellular RNA from undifferentiated, differentiated, and transfected trophoblast cells were isolated using TRIzol reagent (Ambion), as per the manufacture's recommendation. For RNA isolation from undifferentiated TS cells, cells were trypsinized using 0.05% trypsin to avoid giant cell contamination. Cell pellets obtained following trypsinization were washed with PBS and lysed with TRIzol. For differentiated trophoblast cells, cells were directly lysed with TRIzol after washing them with PBS.

First-strand cDNA was synthesized from 5 μg of total RNA with oligo-dT primer using M-MLV Reverse Transcription kit (Invitrogen). cDNAs were then subjected to real-time PCR amplification for various trophoblast-related genes, as described previously (Saha et al, 2017; Chakraborty et al, 2018).

Tenfold dilution of cDNAs from every sample was used for real-time qPCR. PCR reactions were run using Power SYBR Green PCR Master Mix (Applied Biosystems) using the 7500 real-time PCR system (Applied Biosystems) with the following thermal condition: initial holding at 95°C for 10 min followed by 40 cycles of 95°C for 15 s and 60°C for 1 min and a dissociation stage of 95°C for 15 s, 60°C for 1 min, and then 95°C for 30 s. Primers specific for genes of interest are listed in Table S4. Expression of *Rpl7* RNA was used as an endogenous control. The amount of RNA was analyzed using the standard $2^{-\Delta\Delta Ct}$ relative expression method. At least three different biological replicates were used for every reaction.

### MiRNome PCR array for miRNA expression profiling

A large-scale quantitative real-time miRNome miRNA PCR array (Cat. no. MIMM-216Z; V16.0, 96-well; SABiosciences-Qiagen) was used to investigate the miRNA expression profile in undifferentiated TS cells

and differentiated trophoblast cells. This array format consisted of 12 different PCR array plates for profiling expression of the 940 most abundantly expressed and best characterized miRNA sequences in the mouse miRNA genome (miRNome) as annotated in miRBase release 16 (www.mirbase.org). Total RNA was isolated from TS cells and day 6 differentiated trophoblast cells by using a miRNeasy Mini Kit, as per the manufacturer's protocol. The concentration and quality of the RNA were measured using a NanoDrop 2000 spectrophotometer (Thermo Fisher Scientific) followed by fractionation on a formaldehyde gel. Then, 2 µg of total RNA from each sample was reverse transcribed using a miScript-II RT kit, which facilitates the selective conversion of mature miRNAs to cDNAs. These cDNA were used to perform mouse miRNome miRNA PCR array using a miScript SYBR Green PCR Kit according to the manufacturer's instructions. Normalization was performed using the arithmetic mean of three small RNAs, SNORD68, SNORD95, and SNORD96A, which showed very little or no change in stem cell and differentiated cells. All reagents were obtained from Qiagen. SABiosciences miRNA array analysis software was used for data analysis and to calculate the fold change in miRNA expression. Only those miRNAs that met the recommended cutoff readings (Ct ≤ 30) in at least one of the two groups were considered for analysis.

### TaqMan assay

TaqMan assays were performed as described previously (Saha et al, 2017) for validating the expression pattern of the members of two miRNA clusters, miR-290 cluster and miR-503 cluster, which showed differential expression in TS cells and differentiated trophoblast cells in the PCR array. Total RNA was isolated from TS cells and day 6 differentiated trophoblast cells using a miRVana RNA isolation kit (Ambion). Then, 50 ng of total RNA from every sample was reverse transcribed using specific RT primers for selected members of miR-290 and miR-322 clusters (miR-291a-5p, Assay ID 001202; miR-291b-3p, Assay ID 002538; miR-292a-3p, Assay ID 002593; miR-294-3p, Assay ID 001056; miR-295-3p, Assay ID 000189; miR- 322-5p, Assay ID 001076; miR-503-5p, Assay ID 002456; miR-351a-5p, Assay ID 001067; miR-542-3p, Assay ID 001284; miR-450b-5p, Assay ID 001962 and U6, Assay ID 001973), using a TaqMan MicroRNA RT Kit (Applied Biosystems). Expression of each miRNA was determined by TaqMan assay using specific TaqMan probes and TaqMan Universal PCR Master Mix (Applied Biosystems) with a standard thermal cycling condition which includes initial denaturation at 95°C for 10 min, followed by 40 cycles of denaturation for 15 s at 95°C, annealing, and extension for 1 min at 60°C. The miRNA levels were normalized to the U6 snRNA expression level. Samples were analyzed in triplicates from minimum three biological replicates. The amount of miRNA was normalized relative to the amount of U6 snRNA by using the $2^{-\Delta\Delta Ct}$ method.

### Western blot analysis

Western blot analysis was performed as previously described (Chakraborty & Ain, 2017). Total protein was extracted from trophoblast cells using RIPA buffer (20 mM Tris–HCl, pH 7.5, 150 mM NaCl, 1 mM Na2EDTA, 1 mM EGTA, 1% NP40, 1% sodium deoxycholate, 2.5 mM sodium pyrophosphate, 1 mM β-glycerophosphate, 0.2 mM PMSF, and 1 mM sodium orthovanadate) containing Protease

Inhibitor Cocktail (Sigma-Aldrich). A Bio-Rad protein assay reagent (Bio-Rad) was used to estimate the concentration of each protein sample. Then, 80–100 µg of protein extracts were fractionated using 10–12% SDS–PAGE under reducing condition and were then transferred onto to PVDF membranes (Millipore). The membranes were then blocked with 5% skim milk in TBS-T for 1 h at room temperature. Primary antibodies were diluted in milk or BSA as per the manufacturer's instructions and incubated with the membranes at 4°C overnight. Secondary antibodies were diluted in TBS-T and incubated with the membranes at room temperature for 1 h 30 min. An ECL reagent, Luminata Forte (Millipore), was used for chemiluminescence signal detection. Images were acquired with the BioSpectrum 810 imaging system (UVP), and band intensities were quantified using NIH ImageJ software (https://imagej.nih.gov/ij/) and normalized to RPL7 for each sample. Each experiment was performed in triplicates using different biological replicates.

### Antibodies

Antibodies used for this study were purchased from Cell Signaling Technology, Santa Cruz Biotechnology, and Abcam. Primary antibodies purchased from Cell Signaling Technology were as follows: anti-P27 (Cat. no. 2552, dilution used 1:250), anti-RBL2 (Cat. no. 13610, dilution used 1:1,000), anti-cyclin D1 (Cat. no. 2922, dilution used 1:250), and anti-CDC25B (Cat. no. 9525, dilution used 1:1,000). Antibodies obtained from Santa Cruz Biotechnology were anti-P21 (Cat. no. sc-397, dilution used 1: 100), anti-WEE1 (Cat. no. sc-5285, dilution used 1:100), and anti-cyclin E1 (Cat. no. sc-247, dilution used 1:100). Anti-CDX2 (Cat. no. ab88129) and anti-E2F7 (Cat. no. ab56022) antibodies were purchased from Abcam, and both were used in 1:1,000 dilutions. Anti-PLF antibody was obtained from R&D Systems (Cat. no. AF1623) and used in 1:1,000 dilution. HRP-conjugated goat anti-rabbit (Cat. no. 7074) and horse anti-mouse IgG (Cat. no. 7076) antibodies were purchased from Cell Signaling Technology and used in 1:2,000 dilutions. HRP-conjugated donkey anti-goat IgG (Cat. no. A50-101P) was purchased from Bethyl Laboratories and used in 1:10,000 dilution. For immunofluorescence, the same anti-RBL2, anti-cyclin D1, and anti-CDX2 were used in aforementioned dilutions. Anti-BrdU antibody (Cat. no. 94079; Cell Signaling Technology) was used in 1:100 dilution to immunostain the cells. All the fluorochrome-conjugated secondary antibodies used for immunofluorescence were purchased from Sigma-Aldrich. They were FITC-conjugated goat anti-rabbit IgG (Cat. no. F0382), FITC-conjugated goat anti-mouse IgG (Cat. no. F2012), and TRITC-conjugated goat anti-rabbit IgG (Cat. no. T6778). All of these secondary antibodies were used in 1: 2,000 dilutions.

### RNA interference and transient transfection

Transient transfection, overexpression (using p3XFLAG-CMV-10 vector, Cat. no. E7658; Sigma-Aldrich), and RNA interference for *Cdx2* transcripts were carried out as described previously (Chakraborty & Ain, 2017). Transfection of mimics and inhibitors of miRNAs were performed using previously published protocols (Saha et al, 2015, 2017). All siRNAs, mimics, and inhibitors for miRNAs used in this study were purchased from Ambion. TS cells were plated in a 35-mm dish 24 h before transfection. Cells at 60–70% confluence were transfected with combinations of two Cdx2-siRNAs

at a final concentration of 100 nM each (titrated for maximum down-regulation) using Lipofectamine RNAiMAX (Invitrogen) as per the manufacturer's instructions. Similarly, mimics and inhibitors for specific miRNAs were transfected at final concentrations of 200 and 300 nM, respectively. To overexpress or inhibit the whole miRNA clusters (miR-290 or miR-322 cluster), mimics or inhibitors were transfected in combination of all the mimics or all the inhibitors for the respective clusters at final concentrations of 50 nM for each mimic or inhibitor. For every transfection, cells were incubated with transfection mix for 6 h in 37°C in a 5% $CO_2$ incubator, and then, cells were maintained in stemness-maintaining conditions (MEF-CM+FGF4+heparin) for next 48 h. RNA and protein were isolated from transfected cells 48 h after transfection. Cells transfected with scrambled siRNA or mimic or inhibitor were used as control.

## BrdU cell proliferation assay

BrdU cell proliferation assay was performed using a kit from Cell Signaling Technology. TS cells were transfected either with all the inhibitors of miR-290 cluster members or with all the mimics of miR-322 cluster members. Each mimic or inhibitor was transfected at a final concentration of 50 nM. Following 16 h of transfection, the cells were trypsinized and seeded in 96-well plates in triplicates (5 × $10^3$ and 10 × $10^3$ cells/well) and cultured for 48 h in TS media supplemented with BrdU solution. At the end of the incubation period, the BrdU containing medium was removed, and the cells were incubated with 100 µl of fixing/denaturing solution at room temperature for 30 min. After removal of the solution, the cells were incubated with 100 µl of BrdU detection antibody (1:100) for 1 h. The cells were then washed with wash buffer three times and incubated with 100 µl of HRP-linked anti-mouse IgG (1:100) for 30 min. The cells were then washed three times with wash buffer and incubated with 100 µl of TMB substrate for another 30 min. Depending on the color change, 100 µl stop solution was added, and the absorbance was measured at 450 nm using a multimode reader (Perkin Elmer).

## Immunofluorescence

Immunofluorescence staining of antigens was performed following the protocols described previously (Chakraborty et al, 2018). Briefly, TS cells were transfected either with mimic or inhibitor for all the members of the miR-290 cluster or miR-322 cluster as per the requirement. Each mimic or inhibitor was transfected at a final concentration of 50 nM. 24 h following transfection, the cells were trypsinized and seeded onto glass cover slips in 35-mm culture plates and cultured for an additional 24 h. Cells were then fixed with 4% paraformaldehyde (Sigma-Aldrich) for 15 min at room temperature followed by washing with 1× PBS for three times and blocked with blocking solution (1× PBS, 5% serum and 0.3% Triton X-100) for an hour. Subsequently, cells were incubated with primary antibodies, specific for different proteins in antibody dilution buffer (1× PBS, 1% BSA, and 0.3% Triton X-100) and incubated for 2 h at room temperature. Following washing with PBS, cells were incubated with FITC- or TRITC-conjugated anti-rabbit or anti-mouse IgG in antibody dilution buffer for 2 h at room temperature in dark. Cells were then washed with 1× PBS three times and counterstained with Hoechst (2 µg/ml) 1× PBS for 20 min in dark. Cells were washed with 1× PBS for five to six times, and the cover slips were mounted onto glass slides using

Fluoroshield mounting medium (Sigma-Aldrich). Stained cells were imaged at 200× magnification using a Leica DMi8 epifluorescence microscope.

To assess BrdU immunofluorescence, following transfection cells were seeded onto the coverslip in 35 mm plates. BrdU incorporation was performed as per the protocols described earlier. The secondary antibody used for detection was FITC-conjugated anti-mouse IgG.

Fluorescence intensity of each cell and background fluorescence were measured using NIH ImageJ. Quantitative fluorescence intensity was expressed as corrected total cell fluorescence [CTCF = Integrated Density of each cell – (Area of selected cell × Mean fluorescence of background readings)]. CTCF of cells from five different microscopic fields was calculated and normalized against the cell number (CTCF/cell). Three biological replicates were used. CTCF/cell is expressed as mean ± SEM. *$P$ < 0.05; **$P$ < 0.005; ***$P$ < 0.0005; ns, nonsignificant when compared with scrambled transfected control.

## Dual luciferase assay

Murine Cdx2 3'UTR 573-nt fragment (243–815 bp) containing the binding site for miR-542-3p and miR-322-5p/503-5p (5'-CTGTCAC-3' and 5'-TGCTGCT-3', respectively) was cloned in pmirGLO vector (Promega) downstream of the firefly luciferase gene to generate "wild-type" luciferase reporter plasmid. The mutated version of Cdx2 3'UTR was generated by using a Phusion Site-Directed Mutagenesis Kit (Thermo Fisher Scientific) according to the manufacturer's instructions. This mutagenic reaction introduced two point mutations in each of the binding site of miR-542-3p (5'-CTATCGC-3') and miR-322-5p/503-5p (5'-TCCTGAT-3').

Dual luciferase assay was performed as described previously (Saha et al, 2015, 2017). Briefly, HEK-293 cells were plated at 7 × $10^4$ cells/well in 96-well plates 24 h before transfection. For control samples, cells were co-transfected with 150 ng pmirGLO reporter plasmid (Promega) containing either wild-type or mutated 3'UTR along with 75 nM scrambled mimic. For test samples, cells were transfected with 150 ng reporter plasmid along with 75 nM miRNA-mimic specific for miR-542-3p/322-5p/503-5p. Lipofectamine LTX and Lipofectamine RNAiMAX (Invitrogen) were used for transfection of plasmid DNA and miRNA mimics, respectively. 24 h following transfection, firefly (FL) and renilla luciferase (RL) signals were measured using a Dual-Luciferase Reporter Assay kit (Promega) as per the manufacturer's protocol using a multimode plate reader (Perkin Elmer). Relative luciferase activities were determined by normalizing FL to the RL activity. All experiments were performed in triplicates using at least four different biological replicates, and the data were presented as the mean ± SEM.

## Chromatin immunoprecipitation assay

Chromatin immunoprecipitation (ChIP) assay was performed by using a Simple Chip Enzymatic Chromatin IP Kit (Cell Signaling Technology) as per the manufacturer's instructions. Briefly, TS cells were fixed with 1% formaldehyde for 10 min at room temperature to cross-link proteins to DNA. Over cross-linking was blocked by adding glycine and incubating the cells for 5 min at room temperature. Cells were then washed twice with ice-cold PBS-

containing 1 mM PMSF (Sigma-Aldrich), following which cells were scraped from the culture dishes and centrifuged at 214 RCF for 5 min at 4°C. Cell pellets were subsequently incubated in ice-cold lysis buffer (1× buffer A, 0.5 mM DTT, 1× Protease Inhibitor Cocktail, 1 mM PMSF) for 10 min on ice. Cell nuclei were pelleted down by centrifugation at 855 RCF for 5 min at 4°C. The nuclear pellets were resuspended in ice-cold 1× buffer B supplemented with 0.5 mM DTT, and the suspension was then allowed to digest enzymatically by incubating with micrococcal nuclease for 20 min at 37°C to make genomic DNA fragments of 150–900 bp. After centrifugation at 16,060 RCF for 1 min at 4°C, the nuclear pellet was resuspended in ChIP buffer (1× ChIP buffer, 1× Protease Inhibitor Cocktail, 1 mM PMSF) and incubated for 10 min on ice. To break the nuclear membrane, sonication was performed, and the lysate was centrifuged at 9,503 RCF for 10 min at 4°C to extract the cross-linked chromatin fragments. Subsequently, quantification of chromatin DNA was performed as per the kit protocol, and 10 μg of chromatin DNA was incubated with rabbit anti-CDX2 (Abcam) or normal rabbit IgG (Cell Signaling Technology) overnight at 4°C. Protein G-magnetic beads were added to each IP samples and incubated for 2 h at 4°C. The bead–Ab–chromatin complex was rinsed successively with low-salt wash buffer (1× ChIP buffer) three times and with high-salt wash buffer (1× ChIP buffer, 0.35 M NaCl) once. Finally, the immune complex was reverse cross-linked by heating at 65°C with ChIP elution buffer for 30 min. Protein was digested by proteinase K at 65°C for 2 h. The extracted DNA was purified by using a spin column and subjected to PCR amplification using primers specific for the CDX2-binding site on the murine miR-290 cluster and cyclin D1 promoter/enhancer regions. A primer pair used as negative control for each of the miR-290 cluster and cyclin D1 promoter/enhancer region at a location that does not contain any CDX2-binding site. To confirm whether these amplified CDX2 sites are transcriptionally active, similar ChIP analysis was also performed with anti-RNA polymerase II antibody (Cell Signaling Technology).

Primers used for the ChIP assay are as follows: For the miR-290 cluster promoter/enhancer region, binding site 1 (BS1) fwd: 5′-TTCAAACGAAA-GAATAAACTGAACC-3′, rev: 5′-GAGGCTGGAGGATCTTTGTT-3′. Binding site 2 (BS2) fwd: TTCTACATTTTTAACCCTAGGTGCTTT-3′, rev: 5′-AGCACTGAGT-TCTGTCTTA CCTCTTG-3′, binding site 3 (BS3) fwd: 5′-CCTGCGACCCCC-TAATCA-3′, rev: 5′-GGCCTCAAACTTGTCTACTA CAGAAA-3′. Negative control fwd: TGGCCTACCCATTGCTGGAA-3′. Rev: 5′-GCATATTTAATATACTTTAG-3′. For cyclin D1 promoter/enhancer region, binding site 1 (BS1) fwd: 5′-CCAGC-GAGGAGGAATAGATG-3′, rev: 5′-ACTGGGGTGGTTGCAAAG-3′. Binding site 2 (BS2) fwd: 5′-GGAAGGAGCCTATCGTGTCTC-3′, rev: 5′-GGGTGGGATCT-GAGATTTGTC-3′. Negative control fwd: 5′-GTCACTGACAAATAAGCG-3′, rev: 5′-AAGAACTTGGATTTTTATT-3′.

## Target gene prediction

To predict the target genes of validated miRNAs, three online web resources for miRNA target prediction were used. TargetScan (http://www.targetScan.org), Pictar (http://pictar.mdc-berlin.de), and microRNA.org-Targets and Expression (http://www.microrna.org/microrna/home.do) were used for comparative analysis of the predicted genes.

## Statistical analysis

Each experiment was repeated at least three times using different biological samples. All statistical analyses were carried out using GraphPad Prism 5 software. Statistical significance was analyzed by the $t$ test and ANOVA followed by Newman–Keuls multiple comparison test. $P < 0.05$ was considered as significant and is marked with asterisk(s) in the figures.

## Supplementary Information

## Acknowledgements

We thank Professor Janet Rossant, Hospital for Sick Children, Toronto, Canada, for providing the mouse TS cell line. This work was supported by grant from Science and Engineering Research Board, government of India to R Ain (EMR/2015/001195).

## Author Contributions

S Saha: conceptualization, data curation, validation, investigation, visualization, methodology, and writing—original draft.
R Ain: conceptualization, resources, data curation, formal analysis, supervision, funding acquisition, project administration, and writing—review and editing.

## Conflict of Interest Statement

The authors declare that there is no conflict of interest that would prejudice the impartiality of this scientific work.

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
