## [Reviewer comments · Life Science Alliance]

Life Science Alliance

MicroRNA regulation of murine trophoblast stem cell self-renewal and differentiation

Sarbani Saha and Rupasri Ain

DOI: <https://doi.org/10.26508/lsa.202000674>

Corresponding author(s): *Dr. Rupasri Ain (CSIR-Indian Institute of Chemical Biology)*

Review Timeline:

Submission Date:	2020-02-11
Editorial Decision:	2020-03-05
Revision Received:	2020-08-04
Editorial Decision:	2020-08-20
Revision Received:	2020-08-25
Accepted:	2020-08-31

Scientific Editor: Shachi Bhatt

Transaction Report:

March 5, 2020

Re: Life Science Alliance manuscript #LSA-2020-00674-T

Dear Dr. Ain,

Thank you for submitting your manuscript entitled "MicroRNA regulation of murine trophoblast stem cell self-renewal and differentiation" to Life Science Alliance. The manuscript was assessed by expert reviewers, whose comments are appended to this letter.

As you will see, the reviewers appreciate your findings and provide constructive input on how to further strengthen your manuscript. We would thus like to invite you to submit a revised version to us, addressing the individual reviewer points. This seems rather straightforward, but please do get in touch in case you would like to discuss individual points further.

Thank you for this interesting contribution to Life Science Alliance. We are looking forward to receiving your revised manuscript.

Sincerely,

B. MANUSCRIPT ORGANIZATION AND FORMATTING:

Reviewer #1 (Comments to the Authors (Required)):

REVIEW OF MANUSCRIPT

"MicroRNA regulation of murine trophoblast stem cell self-renewal and differentiation"

Saha, S. & Ain, R.

The authors have conducted the current study to elucidate the role of microRNAs in trophoblast development. Using an in vitro mouse trophoblast model, the authors have profiled the miRNA expression signature of trophoblast stem cells (TSCs), and TSCs forced to differentiate (dTR) by the accepted conventional method of withdrawing Fgf4, heparin and feeder-cell conditioned medium. Amongst other differentially expressed miRNAs, this analysis identified 2 miRNA clusters, one enriched in TSCs (mir-290 cluster) and the other enriched in dTR (mir-322 cluster). Through

miRNA knockdown and overexpression methods, the authors validate cell cycle repressors as targets of the mir-290 cluster, and cell cycle activators as targets of the mir-322 cluster. The authors then demonstrate that inhibition of the mir-290, or activation of the mir-322 cluster both drive the differentiation of TSCs. Further, they describe a mechanism by which these miRNAs maintain self-renewal or drive differentiation by investigating the key TSC-maintenance transcription factor, Cdx2. They were able to demonstrate that Cdx2 positively regulate mir-290 expression to maintain the stem cell fate, but that mir-322 can sufficiently downregulate CDx2 levels and result in differentiation of TSCs.

COMMENTS, CRITICISMS AND CONCERNS

Overall, the investigative approach of this study is valid, and the study is technically sound. The claims are supported by the experimental data. The authors have done a thorough job of investigating the mechanism, and I commend the utilization of both overexpression and knockdown methods in the study.

OVERALL COMMENTS ABOUT MANUSCRIPT AESTHETICS

- 1) The manuscript contains many spelling and grammatical errors that should be revised to improve clarity.
- 2) Background information presented in the introduction is appropriate and concise with good use of the literature, although the reviewer believes the introduction would benefit from an explanation of how mouse TSCs are derived and maintained in vitro, how they can be induced to differentiate, and what populations typically arise from in vitro differentiation (we cannot get the full repertoire of differentiated trophoblast cells from simply withdrawing Fgf4 and feeders). Also, a brief description of miRNA biogenesis may be warranted as the authors mention the Dicer knockout study (Spruce et al) as evidence that miRNAs are required for trophoblast stemness but do not explain to the reader the role of Dicer in miRNA biogenesis.
- 3) I am not sure that the list of transcription factors expressed by TSCs and dTR is necessary, it takes up a lot of space. Rather, list 1 or 2 factors (Cdx2 surely because it is investigated in the study) and utilize a reference to draw the readers' attention to more.
- 4) Correct the reference (author's name) on Page 5 (it should read Morales-Prieto).
- 5) Avoid using the term "overexpression" when you mean the protein levels increased as a result of treatment. Overexpression is reserved for artificially induced expression of a gene.

RESULTS

The authors do a nice job at describing the role of miRNAs mechanistically however, do not adequately characterize the resulting dTR phenotype. Does differentiation by inhibition of mir-290 differ from differentiation by Fgf4 removal, versus differentiation by overexpression of mir-322? Are certain types of differentiated trophoblast cells more predominant in one method versus another?

Line 18 The sub-heading says miRNA microarray but the authors used PCR arrays- these are not the same as microarrays, please revise.

It is not clear from the study whether TSCs were cultured in stemness-maintaining conditions (+Fgf4) or differentiation conditions (-Fgf4) following transfection of mimics and inhibitors. This

information is important and should be mentioned in the results as well as elaborated on in the Methods section.

Figure 2 Scramble or mir-290 mimic treatment should have no effect on stem-ness, but mir-290 cluster inhibition should drive differentiation, however, I only see TSC colonies in the mir-290 inhibited group and not the mimic or scramble, which is perplexing.

Figure 3 Similarly in the bright field morphology images of Figures 3, the mir-322 mimic treated image looks like it actually contains TSC colonies yet these colonies are not expressing Cyclin D1, whereas the scramble and 322-inhibitor which I would expect to see TSC colonies in have none.

Figure 4 mir-290 inhibitor or mir-322 mimic treatment should cause differentiation, yet both these panels show colonies of TSCs whereas the scramble control does not. Interestingly these colonies do not stain positive for Cdx2. Brdu positive cells should also be positive for Cdx2 as these should be cycling and we assume stem cells, can the authors describe the Brdu positive-Cdx2 negative cells?

Can the authors please include a representative image of what they are considering a TSC culture and what their day-6 differentiated trophoblast culture looks like?

Although this reviewer does not think this is necessary, it would be interesting assess expression of mir-290 and mir-322 clusters in a differentiation time course experiment.

1. Does inhibition or activation of cell cycle cause differentiation of TSCs?

METHODS

Overall, there is sufficient detail in the Methods to allow reproducibility.

Antibody information for immunofluorescence is missing. Please include source and dilution used.

DISCUSSION

The Discussion could benefit from some more interpretation and speculation. The results can be further discussed in the context of previous literature.

Do the authors believe that manipulating the cell cycle regulators directly (without interfering with miRNA expression) would drive differentiation of TSCs?

Overall, the reviewer believes that the study is very strong. The authors have done a nice job at teasing apart the mechanism of miRNAs driving differentiation. It is nice that you have tied in miRNA and Cdx2 as well as the cell cycle regulators. Provided that these minor revisions are addressed, this study could be of interest to the field of trophoblast and reproductive biologists and contribute to the new and rapidly evolving field of non-coding RNAs.

Reviewer #2 (Comments to the Authors (Required)):

In the manuscript, "MicroRNA regulation of murine trophoblast stem cell self-renewal and differentiation," the authors evaluate the role of two miRNA families, miR-290 and miR-322, in the regulation of trophoblast stem cell differentiation. Overall, the work presented in this manuscript is novel, interesting, well laid-out, and rigorously performed. I only have some relatively minor comments;

- They should show the individual data points for bar graphs
- There should be quantification of IF for CDX2, CYCLIN D1, and RBL2
- I find the second paragraph of the discussion confusing and mostly irrelevant to the story. I recommend removing.
- Since CDX2 often binds with ELF5 and EOMES to form a core transcriptional network in mouse TSCs, I am curious if these other factors have binding sites in the miR290 promoter regions, or have miR322 seed sequences in their 3'UTRs

Reviewer #3 (Comments to the Authors (Required)):

In this manuscript, the authors reported novel functions of two micro RNA (miRNA) clusters on the regulation of self-renewal of mouse trophoblast stem (TS) cells. They first screened miRNAs for their differential expressions in undifferentiated and differentiated TS cells and found that miR290 clusters express at higher levels in TS cells than differentiated derivatives and miR322 cluster show opposite pattern. Then they analyzed their functions and revealed that miR290 cluster targets a set of cell-cycle inhibitors whereas miR322 cluster inhibits a set of cell cycle activators. Both miR290 inhibition and miR322 activation resulted in induction of differentiation of TS cells with down-regulation of TS markers such as Cdx2, Eomes and Esrrb. Moreover, they presented that the TS-specific transcription factor Cdx2 activate the transcription of miR290 to maintain TS self-renewal whereas miR322 inhibits Cdx2 expression, suggesting that they make a system to control self-renewal and differentiation of TS cells.

It was reported that miRNA modulates the TS cell state. For example, Nosi et al reported that mir15, miR322 and miR467 express at higher levels in TS cells than in ES cells and their overexpression in ES cells result in their trans-differentiation to TS cells (Cell Rep, 2017). The role of miR322 in TS self-renewal is also mentioned in this study, but it could be controversial to the previous report since the present one demonstrate the role of miR322 to induce differentiation of TS cells whereas the previous one showed its role to establish TS cells from ES cells. If the activation of miR322 acts in both steps, the continuous overexpression of miR322 will cause the trans-differentiation of ES cells to TS cells and then induce their differentiation. However, the previous report showed the successful capture of miR322-overexpressing ES cells at TS cell state. Therefore, clear explanation on this point will be required in this manuscript. In addition, there are several points required revision for publication.

1. The point described above.
2. The authors hypothesize that the modulation of Cdx2 expression by the two miRNAs is important to switch the status of TS cells from self-renewal to differentiation. If this is the case, it might be possible to overcome the effect of the inhibition of miR290 or activation of miR322 by the overexpression of Cdx2 in TS cells. This point should be addressed to make a clear answer.
3. Do the effect of inhibition of miR290 is counteracted by inhibition of miR322, or vice versa?
4. Fig2c, Fig3c and Fig 4b: If the authors want to make the argument in quantitative manner, these immunostaining data should be converted into quantitative manner.

5. Page 14 line 16: The authors emphasized the roles of the FGF4 signal and Cdx2 to maintain self-renewal of TS cells. However, there is no direct connection between them. It was reported that the response of Cdx2 to the FGF4/MAPK signal is not so sharp. Instead of it, Sox2 and Esrrb respond to the signal very rapidly and they are verified as functional targets to support self-renewal of TS cells (Adachi et al, Mol Cell, 2013).

Response to reviewers

At the outset authors thank the editor for extending the time for review. Authors are grateful to all three reviewers for critical advice. We have taken note of each concern raised by reviewers and have done experiments as per their advice and incorporated the revised data in the manuscript.

- Figures 2, 3 and 4 have been modified by a) replacement of 2C, 3C, 4B with new images showing more TS colonies as per reviewer 1's advice and b) inclusion of 2D, 3D and 4C showing "quantification of immunocytochemistry data".
- Supplementary figure numbers have been changed due to incorporation of new supplementary figures S1, S4 and S7.
- All bar graphs in supplementary have been made in colour.
- Previous figure S1 has been changed to S2.
- Previous figure S2 has been changed to S3.
- Previous figure S3 has been changed to S5.
- Previous figure S4 has been changed to S6.
- Previous figure S5 has been changed to S8.
- All the supplementary tables are figures are now marked with "S" replacing "EV".

The new incorporations in the text of the manuscript have been marked red.

Reviewer 1:

1) Spelling and grammar have been checked thoroughly as per reviewer's comment.

2) Reviewer advised that the introduction would benefit from an explanation of how mouse TSCs are derived and maintained in vitro, how they can be induced to differentiate, and what populations typically arise from in vitro differentiation.

The only available mouse TS cell line derived in Dr. Janet Rossant's laboratory has been used for the studies described in this manuscript. Derivation and maintenance of this cell line has been elaborated in her publication, which has been cited (Tanaka et al., 1998). This has been mentioned in the a) Introduction section (Pg. 4, lines 1-2) and b) Materials and Methods section (Pg. 20). Besides, we have

introduced the morphology of these cells (TS and differentiated cells) in Fig. S1A and incorporated lines 11-15 pg 20 in the Materials and Methods section.

The reviewer suggested incorporation of a brief description of miRNA biogenesis to in the introduction.

Biogenesis of miRNA has been introduced. Pg. 5, lines 1-18.

3) The reviewer advised to remove the various transcription factors involved in self-renewal and differentiation and use “references” to draw reader’s attention.

This section in the introduction has been modified. Pg. 4, lines 3-10. The paragraph describing cell cycle regulators has now been merged with transcription factors. Pg 4, lines 10-17.

4) The reviewer pointed out the incorrect reference (author's name) on Page 5.

This has been corrected. Pg 6, line 7.

5) The reviewer advised to avoid using the term "overexpression" when the protein levels increased as a result of treatment.

While we appreciate reviewer’s view in this matter, we humbly draw attention of reviewer that usage of “overexpression” for exogenous introduction of miRNA mimics is widely used in literature. A few representative examples are mentioned below (a-c). Therefore, we have not made any changes in this regard.

- a) 1.Hashimoto, K., et.al. (2018) Cancer-secreted hsa-miR-940 induces an osteoblastic phenotype in the bone metastatic microenvironment via targeting ARHGAP1 and FAM134A. *PNAS*. 115, 2204-2209.
- b) Bhinge, A., et.al. (2014) MiR -135b is a direct PAX 6 target and specifies human neuroectoderm by inhibiting TGF - β /BMP signaling. *EMBO J.* 33, 1271-1283.
- c) Chen, Y., et al. (2017) MicroRNA-133 overexpression promotes the therapeutic efficacy of mesenchymal stem cells on acute myocardial infarction. *Stem. Cell. Res. Ther.* 8:268.

6) The reviewer has raised concern regarding characterization of differentiated trophoblast cells resulting from a) miRNA290 cluster inhibition, b) miR322 cluster overexpression and c) withdrawal of mitogens.

New experiments have been done to address this concern of the reviewer and data has been presented in Fig. S4.

Differentiation was induced in TS cells using three different methods as suggested by the reviewer and the differentiated trophoblast cell phenotype was assessed using genetic markers for various lineages. *Plf*, *Pl1* and *Ctsq* were used as markers for trophoblast giant cells, *Mash2* and *Tpbp α* were used as spongiotrophoblast marker and *Gcm1* was used as syncytiotrophoblast marker.

This has been incorporated in A) the result section, Pg 11, lines 23,24 and Pg 12, lines 1-11 and B) the discussion section, Pg 18, lines 10-15.

7) The reviewer suggested revising “miRNA microarray” in Line 18.

“microarray” has been changed to “PCR-array” as suggested by the reviewer in Pg 7, line 2.

8) The reviewer inquired whether TSC s were cultured in stemness-maintaining conditions (+Fgf4) or differentiation conditions (-Fgf4) following transfection of mimics and inhibitors.

Throughout the study TSCs were cultured in stemness-maintaining conditions (+Fgf4) following transfection of mimics and/or inhibitors. This information has been incorporated in the Materials and Methods section under “RNA Interference and Transient Transfection” in Pg 27, line 2.

9) The reviewer raised concern regarding presence of TSC colonies in Scramble or mir-290 mimic treatment in Figure 2.

We have now provided a better image with more TSC colonies in Scramble or mir-290 cluster mimic treatment. It may also be noted that TSC doubling time is approximately 36 h. Transfection is done 24h following plating (trypsinization). Therefore, the size of TSC colonies is smaller in scramble and mir-290 cluster mimic treatment. The flattened out cells in mir-290 cluster inhibitor treatment are

differentiated cells and hence they are bigger, which can be found with careful observation.

10) The reviewer is concerned about the bright field morphology images in Figure 3. The reviewer pointed out that the mir-322 mimic treated image looks like it actually contains TSC colonies yet these colonies are not expressing Cyclin D1. The reviewer also pointed that the scramble and 322-inhibitor has no TSC colonies.

We have now provided a better image with more TSC colonies in Scramble or mir-322 cluster inhibitor treatment. Doubling time of TSCs along with plating time contributes to small size of TSC colonies in these treatments. The bigger and spread out cells seen in mir-322 cluster mimic treatment are differentiated and therefore, they are not expressing CYCLIN D1.

11) The reviewer raised concern regarding TSC colonies shown in mir-290 cluster inhibitor or mir-322 cluster mimic treatment and their expression levels of CDX2 and BrdU incorporation in Figure 4.

Most TSC colonies are present only in scramble treated panel. Cells seen in mir-290 cluster inhibitor or mir-322 cluster mimic treatment are mostly differentiated. Therefore, they do not express CDX2 and BrdU incorporation is very low for this cell population.

12) The reviewer advised to include a representative image of TSC culture and day-6 differentiated trophoblast culture.

This has been incorporated in Fig. S1A.

13) The reviewer suggested assessing expression of mir-290 and mir-322 clusters in a differentiation time course experiment.

We have assessed expression status of two representative miRNAs from each cluster in day2, day4 and day 6 differentiated trophoblast cells. Results are shown in Fig.S1B.

We observed gradual temporal decrease of miR-290 cluster members, miR-291b-3p and miR-295-3p with progression of differentiation. MiR-322 cluster members, miR-322-5p and miR-503-5p levels are expressed at considerably high

levels upon induction of differentiation on day 2 and day 4. However, a robust up regulation was observed on day 6 of differentiation. This has been incorporated in result section, Pg. 8, lines 11-18 and in discussion section, Pg. 17, lines 7-9.

14) The reviewer inquired whether inhibition or activation of cell cycle causes differentiation of TSCs?

Inhibition of cell cycle is required to induce the differentiation of TSC, whereas, activation of cell cycle is necessary for the cell proliferation that lead to TSC stemness maintenance. Available literature in this regard has been mentioned in the Introduction, Pg. 4, lines 11-17.

15) The reviewer pointed out the missing information for source and antibody dilution pertaining to the immunofluorescence data.

This information has now been incorporated in materials and method section under “ Antibodies” in Pg 25, lines 5-24 and Pg 26, lines 1-6.

16) The Discussion could benefit from some more interpretation and speculation. The results can be further discussed in the context of previous literature.

We would like to state humbly that we do appreciate reviewer’s suggestion. However, we believe that we adequately discussed our data and insights derived from the data. More interpretation and speculation without experimental support might be misleading. However, we incorporated discussion pertaining to new experiments done as per reviewers’ suggestions.

17) The reviewer inquired whether manipulating the cell cycle regulators directly (without interfering with miRNA expression) would drive differentiation of TSCs?

Yes, we believe that manipulating the cell cycle regulators directly (without interfering with miRNA expression) would drive differentiation of TSCs. MiRNAs act upstream of cell cycle regulator expression. Therefore, it is likely that manipulation of cell cycle regulators will directly affect TSC differentiation.

Reviewer 2:

1) The reviewer advised to show individual data points for bar graphs.

We have included each data point in "Source data". Therefore, we have not changed the bar graphs.

2) The reviewer asked for quantification of IF for CDX2, CYCLIN D1, and RBL2.

Quantification of IF for CDX2 has been introduced to Figure 4C.

Quantification of IF for CYCLIN D1 has been introduced to Figure 3D.

Quantification of IF for RBL2 has been introduced to Figure 2D.

Method for quantification has been incorporated in Pg 29, lines 3-10.

Figure legends have been changed accordingly on Pg 39, lines 23-24; Pg 40, lines 12-13 and 20-21.

3) The reviewer recommended removal of the second paragraph of the discussion as s/he found the same irrelevant and confusing.

We have removed only a point, which was remotely connected regarding miRNA regulation by PRC complex that act as a sponge in ES cells.

Rest of the points were kept untouched as we think that it gives insights into our data and also keeping in mind that other two reviewers did not object on this.

4) The reviewer wanted to know whether ELF5 and EOMES have binding sites in the miR290 promoter regions, or have miR322 seed sequences in their 3'UTRs

ELF5 and EOMES binding sites are not present in miR290 promoter.

MiR322 seed sequence is not present in the 3'-UTR of ELF5 and EOMES.

Reviewer 3:

1) The reviewer has the following concern: It was reported that miRNA modulates the TS cell state. For example, Nosi et al reported that mir15, miR322 and miR467 express at higher levels in TS cells than in ES cells and their overexpression in ES cells result in their trans-differentiation to TS cells (Cell Rep, 2017).

MiR322 targets cell cycle activators (our data). In real time PCR, the Ct values of miR322 in TS cells and differentiated cells are ~ 22.1 and 17.44, respectively. This means miR322 is expressed in TS cells but its expression increases 16-18 fold upon TS cell differentiation.

ES cells are pluripotent and TS cells are multipotent. Therefore, it is likely that moderate expression of miR322 leading to down regulation of cell cycle activators in ES cells will drive differentiation into default TS pathway. For TS differentiation, robust up regulation of miR322 is required as shown in our manuscript. So, our data do not contradict the previously published result of Nosi et al. (2017). This point has been incorporated in discussion section in Page 19, lines 5-9.

2) The reviewer suggested that if our hypothesis that the modulation of Cdx2 expression by the two miRNA clusters is important to switch the status of TS cells from self-renewal to differentiation whether CDX2 overexpression can overcome A) the effect of miR290 cluster inhibition or B) miR322 cluster overexpression in TS cells.

We have performed two new experiments to address this concern of the reviewer and data from these experiments has been incorporated in Fig. S7.

We have overexpressed CDX2 along with either inhibition of miR290 cluster (A) or overexpression of miR322 cluster (B) in TS cells and maintained the cells in TS media. Genetic markers for TS cells (Eomes, Esrrb) and differentiated trophoblast cells (Plf and Pl1) were used to assess self-renewing state and differentiated state, respectively. Results from these experiments demonstrate that CDX2 overexpression can reverse the effect of miR290 cluster inhibition or miR322 cluster overexpression. This has been incorporated in result section Pg. 13, lines 19-23 and Pg. 14, lines 1-3. Figure legend for this experiment is on Pg 45.

3. The reviewer inquired whether the effect of inhibition of miR290 is counteracted by inhibition of miR322, or vice versa?

We have performed the following new experiments to address this inquiry of the reviewer. We have inhibited 290 cluster, 322 cluster individually and together in TS cells and maintained the cells in TS media. Genetic markers for TS cells (Cdx2, Eomes, Esrrb) were used to assess self-renewing stat.

Inhibition of 290 cluster led to decrease in TS cell markers. Co-inhibition of 290 and 322 cluster did not further decrease the TS cell markers. This might be due to the following reason. 290 cluster inhibition is expected to increase cell cycle inhibitor leading to differentiation wherein 322 cluster is 16-18 fold up regulated, which cannot be countered by miR322 down regulation by exogenous inhibitor.

Inhibition of 322 cluster led to increase in TS cell markers because miR322 cluster directly target CDX2. Hence, expectedly up regulation of CDX2 will increase the other TS markers as well. Co-inhibition of 322 cluster along with 290 led to decrease in TS cell markers countering its effect. This happened because 290 cluster inhibition is expected to increase cell cycle inhibitor leading to differentiation and decrease in stemness markers. Result from this experiment has not been inserted into the manuscript and is shown below.

4. The reviewer suggested quantification of immunostaining data presented in Fig2c, Fig3c and Fig 4b.

As per reviewer's suggestion, we incorporated the quantification data

Quantification of IF for RBL2 (Fig2c) has been introduced to Figure 2D.

Quantification of IF for CYCLIN D1 (Fig3c) has been introduced to Figure 3D.

Quantification of IF for CDX2 (Fig 4b) has been introduced to Figure 4C.

Method for quantification has been incorporated in Pg 29, lines 3-10.

Figure legends have been changed accordingly on Pg 39, lines 23-24; Pg 40, lines 12-13 and 20-21.

5. The reviewer has the following concern: Page 14 line 16: The authors emphasized the roles of the FGF4 signal and Cdx2 to maintain self-renewal of TS cells. However, there is no direct connection between them. It was reported that the response of Cdx2 to the FGF4/MAPK signal is not so sharp. Instead of it, Sox2 and Esrrb respond to the signal very rapidly and they are verified as functional targets to support self-renewal of TS cells (Adachi et al, Mol Cell, 2013).

Without FGF4 stimuli stemness of TS cells cannot be maintained. In absence of FGF4 TS cells spontaneously differentiate. In addition, TS cells in stemness condition expresses huge amount of Cdx2. Down regulation of Cdx2 leads to TS differentiation. Therefore, FGF4 as an external stimuli and CDX2 as a transcription factor are both essential for TS maintenance.

We also would like to draw attention of the reviewer to the following figure that was summarized by Adachi et al (2013) in Mol Cell.

[Figure removed by Life Science Alliance editorial staff per authors' request]

FGF4 directly regulate Sox2,
which is a common regulator
of both ESC and TSC.

However, the cell type specific transcription factor that directly maintains stemness of TSC is CDX2. We agree with the reviewer that Esrr β indeed is directly regulated by FGF4.

August 20, 2020

RE: Life Science Alliance Manuscript #LSA-2020-00674-TR

Author information redacted

Dear Dr. Ain,

Thank you for submitting your revised manuscript entitled "MicroRNA regulation of murine trophoblast stem cell self-renewal and differentiation". We would be happy to publish your paper in Life Science Alliance pending final revisions necessary to meet our formatting guidelines and addressing the minor concerns raised by the reviewer (see comments below).

Along with the points listed below, please also revise the following:

- *DATASET EV LEGENDS: please add tables as excel or doc file
- for fig. S6, there is a mention of panel F in the fig. Legend, but this is not part of the figure and doesn't have a callout
- REFERENCE FORMAT: -please list 10 authors et al.
- please provide Source data

A. FINAL FILES:

B. MANUSCRIPT ORGANIZATION AND FORMATTING:

Sincerely,

Shachi Bhatt
Executive Editor
Life Science Alliance
www.life-science-alliance.org

Reviewer #3 (Comments to the Authors (Required)):

In this revised manuscript, the authors addressed the points raised by the reviewers and made significant revisions. The resulting manuscript was improved and looks appropriate for publication. However, there are minor points required additional revision.

1. In Fig S4, the authors compared the differentiation events of TS cells induced by miR-290 inhibitors, miR-322 mimics and withdrawal of MEF-CM, FGF4 and heparin as the request of Reviewer 1. The result indicated that the induction of differentiation markers induced by miR-290

inhibitors and miR-322 mimics is much weaker than that by withdrawal of MEF-CM, FGF4 and heparin for Ctsq, Mash2 and Gcm1 and Tpbp1 was not well up-regulated. What does this observation mean? Does it mean the bias for the choice of cell type, incomplete differentiation event, or inefficient differentiation? Clear explanation will be required. In addition, although the authors stated that the nature of differentiation induced by either inhibition of miR-290 a cluster or overexpression of miR-322 cluster followed the same pattern of differentiation induced by mitogens withdrawal (page 18 line 10), this is overstatement because of the discrepancy mentioned above.

2. Page 12 line 15: The authors stated 'CDX2 is known to be critical for maintaining stemness of mouse TS cells and its depletion causes spontaneous differentiation of TS cells (Strumpf et al, 2005).' However, Strumpf et al only demonstrated that it is impossible to establish TS cell line from Cdx2-null embryos. The requirement of Cdx2 in ES-derived TS cells was shown by Niwa et al (Cell, 2005). It will be suitable if there is any report showing the knock-out/knock-down of Cdx2 in embryo-derived TS cells.

Response to Editor and reviewers:

- Supplementary tables S1A-S3 have been removed from main text doc as instructed. These tables are now in a separate supplementary doc files.
- Fig. S6 legend has been corrected on page 44, line 21-22.
- References 23 and 26 have been modified as suggested, Pg.36, lines 21-22; page 37, lines 7-8.
- It may be noted that source data is provided.

Response to reviewer

- Reviewer is concerned about the result shown in Fig. S4, which was done as per reviewer 1's suggestion. It may kindly be noted that miR290 cluster inhibition or miR322 cluster overexpression was performed *maintaining TS cells in stemness condition* (mentioned appropriately in the materials and methods section). Hence, there were differences in levels of expression of some differentiation markers when compared with differentiated trophoblast cells obtained by mitogens withdrawal. So we believe that Page 18, line 10 is not an overstatement.
- The reviewer commented on correctness of reference for page 12, line 15. We have changed the text as per Stumpf et al. on page 12, line 15.

August 31, 2020

RE: Life Science Alliance Manuscript #LSA-2020-00674-TRR

Author information redacted

Dear Dr. Ain,

Thank you for submitting your Deleted entitled "MicroRNA regulation of murine trophoblast stem cell self-renewal and differentiation". It is a pleasure to let you know that your manuscript is now accepted for publication in Life Science Alliance. Congratulations on this interesting work.

DISTRIBUTION OF MATERIALS:

Again, congratulations on a very nice paper. I hope you found the review process to be constructive and are pleased with how the manuscript was handled editorially. We look forward to future exciting submissions from your lab.

Sincerely,

Shachi Bhatt, Ph.D.,
Executive Editor
Life Science Alliance

e contact@life-science-alliance.org
www.life-science-alliance.org